# Incidence of admission ionised hypocalcaemia in paediatric major trauma: protocol for a systematic review and meta-analysis

Owen Hibberd [1,2] James Price [1,3] Tim Harris,[2]
Ed Benjamin Graham Barnard [1,4]

¹Emergency and Urgent Care Research in Cambridge (EUReCa), PACE Section, Department of Medicine, Cambridge University, Cambridge, UK
²Queen Mary University of London, Blizard Institute, London, UK
³Department of Research, Audit, Innovation, and Development (RAID), East Anglian Air Ambulance, Norwich, UK
⁴Academic Department of Military Emergency Medicine, Royal Centre for Defence Medicine, Birmingham, UK

**Correspondence to**
Dr Owen Hibberd;
oh296@cam.ac.uk

## ABSTRACT

**Introduction** Hypocalcaemia forms part of the 'diamond of death' in major trauma, alongside hypothermia, acidosis and coagulopathy. In adults, admission hypocalcaemia prior to transfusion is associated with increased mortality, increased blood transfusion requirements and coagulopathy. Data on paediatric major trauma patients are limited. This systematic review and meta-analysis aims to describe and synthesise the available evidence relevant to paediatric trauma, admission hypocalcaemia and outcome.

**Methods and analysis** The Preferred Reporting Items for Systematic Review and Meta-Analysis Protocols guidelines will be used to construct this review. A planned literature search for articles in the English language will be conducted from inception to the date of searches using MEDLINE on the EBSCO platform, CINAHL on the EBSCO platform and Embase on the Ovid platform. The grey literature will also be searched. Both title and abstract screening and full-text screening will be done by two reviewers, with an adjudicating third reviewer. Heterogeneity will be assessed using the I² test, and the risk of bias will be assessed using the ROBINS-I tool. A meta-analysis will be undertaken using ratio measures (OR) and mean differences for measures of effect. When possible, the estimate of effect will be presented along with a CI and a p value.

**Ethical review and dissemination** Ethical review is not required, as no original data will be collected. Results will be disseminated through peer-reviewed publications and at academic conferences.

**PROSPERO registration number** CRD42023425172.

## STRENGTHS AND LIMITATIONS OF THIS STUDY

⇒ The protocol follows the Preferred Reporting Items for Systematic Review and Meta-Analysis Protocols guidelines.
⇒ This is a novel review that addresses an area of uncertainty in the current evidence base surrounding paediatric major trauma through a systematic review and meta-analysis of published data and the grey literature.
⇒ The review methodology is at risk of limitation by publication bias. Where appropriate, this will be assessed using funnel plots.

## BACKGROUND

Major trauma is one of the leading causes of death in children in the UK.[1 2]

A key cause of potentially survivable death from trauma is haemorrhage.[3] Uncontrollable haemorrhage may be related to the injury mechanism itself or as a result of trauma-induced coagulopathy (TIC).[4] TIC is common, occurring in at least a quarter of haemorrhagic deaths, and has a number of proposed pathophysiological mechanisms that generally involve injury and shock, provoking an immunological, endothelial and platelet response.[4] All forms of haemorrhage are further exacerbated by the 'lethal triad' of coagulopathy, hypothermia and acidosis.[5–7] More recently, biochemical abnormalities such as hyperkalaemia and hypocalcaemia have been recognised to contribute to deaths from haemorrhage.[6 8] In particular, calcium's role is important for clot formation, vascular tone and cardiac contractility, with hypocalcaemia contributing to coagulopathy and cardiovascular decompensation.[5 6] As such, the 'lethal triad' is now considered a 'diamond of death' with hypocalcaemia forming a key component of this deleterious combination.[5 6] The early recognition and treatment of these components in the 'diamond of death' are essential for trauma resuscitation.[5 7 9]

### Rationale

The free form of calcium (ionised calcium (iCa)) is the physiologically relevant component of calcium in the blood.[10] iCa is measured on blood gases, which are often taken on arrival for major trauma patients, and there is good agreement between arterial and venous measurements.[11] Blood

gas measurements will also record the pH and lactate, which can affect the availability of iCa.[12 13] Ionised hypocalcaemia (iHypoCa) in major trauma patients is multifactorial.[5–7 14] The infusion of citrated blood products is a recognised cause of hypocalcaemia in trauma due to calcium chelation with citrate.[6 15] There is also emerging evidence in adults that early hypocalcaemia may occur in trauma patients prior to the receipt of blood products containing citrate.[16–19] Potential pathophysiological mechanisms underpinning this include calcium binding by lactate, the intracellular influx of calcium due to ischaemia and reperfusion, impaired calcium homeostasis secondary to trauma and dilution by crystalloid fluid resuscitation.[5–7] A systematic review and meta-analysis, which included a total of 1213 major trauma patients, 18 years or older, with a documented iCa level on admission, explored the incidence and outcomes associated with admission iHypoCa.[20] Studies that involved patients whose calcium concentration may have been confounded by prior blood transfusions were excluded.[20] Overall, the incidence of admission ionised hypocalcaemia (iHypoCa) was 56.2%, and iHypoCa was associated with increased mortality, increased blood transfusion requirements and coagulopathy.[16–20] Evidence of admission iHypoCa and the association with adverse outcomes in adult trauma patients has led to the early measurement and replacement of calcium being recommended in adult trauma guidelines.[21 22] Paediatric major trauma data are limited. Given the different physiology of children compared with adults, children may be more vulnerable to the effects of iHypoCa, and the results of studies involving adult major trauma patients may not be able to be extrapolated to a paediatric cohort.[23] A search of PROSPERO did not find any similar planned systematic reviews or meta-analyses. Moreover, a preliminary search of the literature has found a few heterogeneous studies, which indicate that admission iHypoCa may be less prevalent in children compared with adults.[24–27]

## Aims

The primary aim of this systematic review and meta-analysis is to explore the limited evidence related to the incidence of admission iHypoCa in paediatric major trauma patients. The review also aims to explore whether admission iHypoCa, compared with normocalcaemia, is associated with adverse clinical outcomes.

## METHODS
### Eligibility criteria

This proposed systematic review and meta-analysis will explore the incidence of iHypoCa in paediatric (<18 years old) major trauma patients (Injury Severity Score (ISS) >15) and explore whether admission iHypoCa (iCa <1.16 mmol/L), compared with normocalcaemia (iCa ≥1.16 mmol/L) is associated with a greater incidence of adverse outcomes.[12] An iCa of <1.16 mmol/L was chosen to reflect different levels of hypocalcaemia thresholds

across the literature and facilitate the inclusion of all relevant studies.[24–26] The Population, Intervention, Comparison, Outcomes and Study Design (PICOS) eligibility criteria are detailed in table 1.

### Information sources

A planned literature search for articles in the English language will be conducted from inception to the search date using MEDLINE on the EBSCO platform, CINAHL on the EBSCO platform and Embase on the Ovid platform. The reference lists of all included studies and the grey literature will also be searched.

### Search strategy

The search strategy can be found in online supplemental tables 1-3.

The search will also involve checking reference lists of retrieved articles, conference abstracts and online study results. If the data are incomplete, then the corresponding authors will be contacted for additional information.

### Study records

The search strategy will be undertaken by a trained librarian and information specialist. The combined abstracts from the search strategy will be independently screened by two reviewers to identify studies meeting inclusion criteria; any duplications will be removed manually. For abstracts meeting inclusion criteria, full texts will be retrieved and again independently reviewed against the inclusion and exclusion criteria by two reviewers and an adjudicating third reviewer.

A standardised data sheet (Microsoft Excel for Mac, V.16.72, 2023) will be used to extract data from included studies to facilitate data synthesis and assessment of quality and risk of bias. The extracted data will be independently verified by the second reviewer, and any discrepancies again be adjudicated by the third reviewer.

The following data items will be extracted:
1. Hospital setting.
2. Study type.
3. Country of treatment.
4. Cohort size.
5. ISS.
6. Abbreviated Injury Scale score for injury regions.
7. Ionised hypocalcaemia definitions.
8. Incidence of admission iHypoCa (iCa<1.16 mmol/L).[12]
9. Definitions and presence of coagulopathy
10. The presence of hyperkalaemia (>5.5 mmol/L)[28]
11. The presence of hyperlactataemia (>2.0 mmol/L)[29]
12. Haemodynamic instability (hypotension (based on age-specific Advanced Paediatric Life Support (APLS) values) or elevated Shock Index Paediatric Age-Adjusted (SIPA)).[30–32]
13. Administration of exogenous calcium.
14. Vasoactive medication requirements within the first 24 hours.
15. Total blood product transfusion requirement during the first 24 hours.

**Table 1** PICOS strategy for inclusion and exclusion

| PICOS strategy | Inclusion criteria | Exclusion criteria |
|---|---|---|
| P | Paediatric (<18 years) major trauma patients (injury severity score >15) with a documented iCa level on admission. | iCa level taken after the administration of blood products in the emergency department. |
| I | Hypocalcaemia on admission (iCa <1.16 mmol/L) | N/A |
| C | Normocalcaemia on admission (iCa ≥1.16 mmol/L) | N/A |
| O | Primary outcome include the incidence of admission ionised hypocalcaemia. Secondary outcomes include the association with physiological abnormalities: ► Haemodynamic instability ► Hyperkalaemia ► Hyperlactataemia ► pH abnormalities ► Coagulopathy and adverse outcomes: ► Vasopressor requirement within 24 hours ► Transfusion requirement within 24 hours ► Activation of the major haemorrhage protocol within 24 hours ► Requiring invasive (operative or interventional radiology) intervention within 24 hours ► Hospital length of stay ► Paediatric Intensive Care Unit length of stay ► Early mortality within 24 hours and medium mortality during episode of hospital admission (>24 hours) or within 30 days | N/A |
| S | Clinical trials (randomised and non-randomised), observational studies (cohort and case-controlled) case reports, case series and literature reviews. | Systematic reviews. Opinion articles. |

iCa, ionised calcium; PICOS, Population, Intervention, Comparison, Outcomes, and Study Design.

16. Activation of the major haemorrhage protocol within the first 24 hours.
17. Requirement for invasive (operative or interventional radiology) intervention within 24 hours.
18. Hospital length of stay (LOS) (days).
19. Paediatric Intensive Care Unit (PICU) LOS (days).
20. Early mortality within 24 hours and medium mortality during an episode of hospital admission (>24 hours) or within 30 days.

## Outcomes and prioritisation

The primary outcome of this systematic review and meta-analysis is the overall incidence of admission iHypoCa. Secondary outcomes are the associations with physiological abnormalities and adverse outcomes. Physiological abnormalities are classified dichotomously as the presence of hypotension (based on age-specific APLS values)[30] or elevated SIPA (0–6 years: >1.22, 7–12 years: >1.00 and 13–16 years: >0.90),[31] [32] hyperkalaemia (>5.5 mmol/L)[28] and hyperlactataemia (>2.0 mmol/L).[29] Adverse outcomes are classified dichotomously as the requirement for vasopressors, transfusion, activation of the major haemorrhage protocol or invasive (operative or interventional radiology) intervention in the first 24 hours and mortality within 30 days. Hospital LOS and PICU LOS in days are classified continuously.

Ratio measures (OR) and mean differences will be used for measures of effect. When possible, the estimate of effect will be presented along with a CI and a p value.

## Risk of bias

The risk of bias will be assessed for all included studies. For any randomised controlled trials, the Grading of Recommendations Assessment, Development and Evaluation methodology will be used, and for observational studies, the Risk Of Bias In Non-randomized Studies of Interventions tool will be used.[33] [34]

The risk of publication bias will be assessed with funnel plots as appropriate.[35]

## Data synthesis

The data will be synthesised following Preferred Reporting Items for Systematic Review and Meta-Analysis guidelines. Studies will be assessed clinically (PICO) and methodologically (study design, comparability, outcome ascertainment and risk of bias). Given that current evidence is likely to be limited, the minimum number of studies is two. A preliminary search has identified four studies.[24–27] The $I^2$ test will be conducted to determine if the data are suitable for quantitative synthesis.[36]

Meta-analysis of effect estimates is intended and will be displayed using a forest plot. If there is limited evidence for a prespecified comparison, then the haemodynamic instability and vasopressor PICO groups may be combined. Definitions of hypocalcaemia will also be combined if required, providing values are iCa<1.16 mmol/L. Other elements are unlikely to be suitable as contingencies for a combination. If different effect measures are used, attempts will be made to transform the effect measures for meta-analysis.

A narrative synthesis and summary of effect measures (with the use of box-and-whisker plots) will be conducted if heterogeneity is deemed too substantial across studies to allow for meaningful meta-analysis or if there are major concerns about bias from the three reviewers.

Meta-analysis or narrative synthesis of elements will focus on the incidence of hypocalcaemia in paediatric trauma patients and the trend towards adverse outcomes. Subgroup analysis may be undertaken for severe iHypoCa (iCa<1.0 mmol/L).

### Patient and public involvement

None.

### ETHICS AND DISSEMINATION

Ethical review is not required, as no original data will be collected. Results will be disseminated through peer-reviewed publications and at academic conferences.

**Acknowledgements** The authors would like to acknowledge and thank Catherine Hancox and the Defence Medical Academic Library Team for their assistance with the search strategy.

**Contributors** OH conceptualised the protocol. OH, JP, TH and EBGB all contributed to the design, data interpretation, critical revision and final approval of the protocol.

**Funding** The authors have not declared a specific grant for this research from any funding agency in the public, commercial or not-for-profit sectors.

**Competing interests** None declared.

**Patient and public involvement** Patients and/or the public were not involved in the design, or conduct, or reporting, or dissemination plans of this research.

**Patient consent for publication** Not applicable.

**Provenance and peer review** Not commissioned; externally peer reviewed.

**ORCID iDs**
Owen Hibberd http://orcid.org/0000-0002-3839-1874
James Price http://orcid.org/0000-0002-9643-692X
Ed Benjamin Graham Barnard http://orcid.org/0000-0002-5187-1952

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
