## [Reviewer comments · BMJ Open]

ARTICLE DETAILS

TITLE (PROVISIONAL)	The Incidence of Admission Ionised Hypocalcaemia in Paediatric Major Trauma: Protocol for a Systematic Review and Meta-Analysis
AUTHORS	Hibberd, Owen; Price, James; Harris, T; Barnard, Ed

VERSION 1 – REVIEW

REVIEWER	Mark Lyttle Bristol Royal Hospital for Children, Emergency Department
REVIEW RETURNED	31-Jul-2023

GENERAL COMMENTS	Dear authors, thank you very much for the opportunity to review this protocol, which outlines a planned systematic review and meta-analysis of the incidence of hypocalcaemia in paediatric trauma, and it's association with adverse outcomes. I found your protocol well constructed, thorough, and easy to read. This addresses an issue which is coming more to the fore in paediatric trauma. Therefore whilst I have concerns over the number of articles you will find, I believe this is a worthy effort, and I look forward to the results. I have no suggestions through which to improve your methodology. Eligibility criteria selection, and systematic review type, are always challenging decisions to make in situations where the volume of literature may be low. I applaud the ambition to undertake a meta-analysis, provided there is sufficient homogeneity across studies. Of note, I also agree with the decision to take a narrative approach, and on this occasion to include to observational studies - the methodology for quality and bias assessment is also appropriate. Once again, I wish you all the best in your endeavour in this area. Mark
--

REVIEWER	Angelo Ciaraglia The University of Texas Health Science Center at San Antonio, Surgery
REVIEW RETURNED	07-Oct-2023

GENERAL COMMENTS	Well written study protocol. Look forward to reading the final manuscript of the systematic review. A few minor comments. Please provide further justification/reasoning for selection of iHypoCa <1.16 mmol/L. The study cited by Egi et al appears to be from a wide range of medical and surgical pediatric ICU patients with a heterogenous pathophysiology that may be contributing to the
--

	hypo and hypercalcemia that was identified in this study. Several of the currently published studies involving trauma-related pediatric hypocalcemia that are referenced in this protocol (reference 23-26, specifically) have different levels of hypocalcemia thresholds that are compared (e.g., Ciaraglia et al used 1.00 mmol/L; Gimelraikh et al used <1 (severe hypocalcemia), 1-1.16 (hypocalcemia), >1.16 normocalcemia; Epstein et al used 1.00 mmol/L as threshold; Cornelius et al used total corrected calcium in their retrospective study). Please expand briefly on reasoning for selection of 1.16 threshold for inclusion/exclusion criteria
--	---

VERSION 1 – AUTHOR RESPONSE

Reviewer 1:

I found your protocol well constructed, thorough, and easy to read.

This addresses an issue which is coming more to the fore in paediatric trauma. Therefore whilst I have concerns over the number of articles you will find, I believe this is a worthy effort, and I look forward to the results.

I have no suggestions through which to improve your methodology.

Eligibility criteria selection, and systematic review type, are always challenging decisions to make in situations where the volume of literature may be low.

I applaud the ambition to undertake a meta-analysis, provided there is sufficient homogeneity across studies.

Of note, I also agree with the decision to take a narrative approach, and on this occasion to include to observational studies - the methodology for quality and bias assessment is also appropriate.

Once again, I wish you all the best in your endeavour in this area.

Mark

- Gratefully noted

Reviewer 2:

Please provide further justification/reasoning for selection of iHypoCa <1.16 mmol/L. The study cited by Egi et al appears to be from a wide range of medical and surgical pediatric ICU patients with a heterogenous pathophysiology that may be contributing to the hypo and hypercalcemia that was identified in this study. Several of the currently published studies involving trauma-related pediatric hypocalcemia that are referenced in this protocol (reference 23-26, specifically) have different levels of hypocalcemia thresholds that are compared (e.g., Ciaraglia et al used 1.00 mmol/L; Gimelraikh et al used <1 (severe hypocalcemia), 1-1.16 (hypocalcemia), >1.16 normocalcemia; Epstein et al used 1.00 mmol/L as threshold; Cornelius et al used total corrected calcium in their retrospective study). Please expand briefly on reasoning for selection of 1.16 threshold for inclusion/exclusion criteria

- An iCa of <1.16 mmol/L was chosen to reflect different levels of hypocalcemia thresholds across the literature and facilitate inclusion of all relevant studies.
- Above added to manuscript